# Milk Exosomes: Perspective Agents for Anticancer Drug Delivery

**DOI:** 10.3390/ijms21186646

**Published:** 2020-09-11

**Authors:** Sergey Sedykh, Anna Kuleshova, Georgy Nevinsky

**Affiliations:** 1SB RAS Institute of Chemical Biology and Fundamental Medicine, 630090 Novosibirsk, Russia; niboch@niboch.nsc.ru (A.K.); s.sedykh@nsu.ru (G.N.); 2Faculty of Natural Sciences, Novosibirsk State University, 630090 Novosibirsk, Russia

**Keywords:** milk, exosomes, milk exosomes, drug delivery, cancer, isolation, vesicles

## Abstract

Exosomes are biological nanovesicles that participate in intercellular communication by transferring biologically active chemical compounds (proteins, microRNA, mRNA, DNA, and others). Due to their small size (diameter 40–100 nm) and high biological compatibility, exosomes are promising delivery tools in personalized therapy. Because artificial exosome synthesis methods are not developed yet, the urgent task is to develop an effective and safe way to obtain exosomes from natural sources. Milk is the only exosome-containing biological fluid that is commercially available. In this regard, milk exosomes are unique and promising candidates for new therapeutic approaches to treating various diseases, including cancer. The appearance of side effects during the use of cytotoxic and cytostatic agents is among the main problems in cancer chemotherapy. According to this, the targeted delivery of chemotherapeutic agents can be a potential solution to the toxic effect of chemotherapy. The ability of milk exosomes to carry out biologically active substances to the cell makes them promising tools for oral delivery of chemotherapeutic agents. This review is devoted to the methods of milk exosome isolation, their biological components, and prospects for their use in cancer treatment.

## 1. Introduction

Exosomes are natural extracellular nanovesicles that participate in intercellular communication by carrying biologically active substances like proteins, microRNA, mRNA, DNA, and other molecules [1,2]. The carried compounds can be transported both inside and on the surface of exosomes [3]. Due to their small size (diameter 40–100 nm), exosomes are promising delivery tools in personalized therapy [4]. Because artificial exosome synthesis methods are not developed yet, the urgent task is to effectively and safely obtain exosomes in mass quantities from natural sources. Tumor cell cultures can hardly be considered as a promising source of vesicles for pharmacology applications. The exosome content is shown in various biological fluids: blood plasma, urine, saliva, milk, amniotic fluid, ascites, cerebrospinal fluid, and others [2]. Among them, milk is the only biological liquid containing exosomes that is available on an industrial scale.

Human milk exosomes were first described in 2007. To date, exosomes have been isolated (chronologically) from human [5], bovine [6], porcine [7], wallaby [8], camel [9], rat [10], horse [11], panda [12], yak [13], sheep [14], and goat milk [15]. Databases provide more than 100 articles on exosomes and other milk vesicles (more than half of them are on human and cow milk vesicles). Large volumes of milk can be obtained from a cow and relatively large from horse, sheep, goat, and camel. Due to possible prion content, bovine milk should not be considered a source of protein-containing structures [16].

Milk exosomes are promising candidates in developing new therapeutic approaches to the treatment of various diseases, including cancer. However, currently, there is only incomplete data in the literature on the use of milk vesicles, particularly exosomes, to deliver biologically active molecules to cells [17]. In the application of milk exosomes for cancer treatment, two directions can be highlighted: the delivery of antitumor drugs [18] and therapeutic nucleic acids [19,20]. Various side effects occurring during the use of cytotoxic and cytostatic agents are unresolved problems of cancer chemotherapy [21]. Cells with a high mitotic index are most susceptible to the action of cytostatic agents; thus, the use of cytostatic drugs leads to the death of not only the tumor cells but also cells of the bone marrow, skin, hair, epithelium of the gastrointestinal tract, etc. [22]. Targeted chemotherapeutic agent delivery can be a potential solution to the occurrence of toxic effects in chemotherapy [4,23].

Because exosomes contain mRNA, microRNA, and proteins that may harm cells, some experts disagree that exosomes (including milk exosomes) can be used as delivery vehicles. Analysis of literature data shows that many papers devoted to the protein and nucleic components of exosomes may have overestimated the number of proteins and/or nucleic acid molecules in exosomes due to contamination of the preparations with co-isolating molecules [24,25]. Therefore, it is essential to analyze highly purified exosome preparations to identify biopolymers that are intrinsic components of exosomes and the proteins and nucleic acids that co-isolate with exosomes. Exosomes can be used as drug delivery systems, carrying the drugs of therapeutic nucleic acids on the surface or after “loading” of these components using electroporation, sonication, or ligands on the surface of exosomes [26]. Another method is the fusion of exosomes with liposomes, loaded with necessary contents to form “chimeric” exosomes [27]. We propose that the use of highly purified exosomes as drug delivery vehicles is critical to avoid unwanted side effects. This review is devoted to the methods of milk exosome isolation, analysis of their biological components, and prospects for their use in cancer treatment.

## 2. Isolation of Milk Exosomes

Various physical, physicochemical, and immunological approaches are used to isolate exosomes from milk. General issues concerning the isolation and characterization of exosomes are described in the guidelines of the International Society for Extracellular Vesicles [28,29]. Because the volume of milk obtained from single humans or animals may vary from milliliters to liters, the general protocol [1] of exosome isolation from milk samples usually starts with a series of centrifugations [6]: first, several sequential centrifugations at low speed of 600–1000× *g* to defat the milk and precipitate the cells, then 10,000–16,000× *g* to precipitate milk proteins, followed by one or several ultracentrifugations at 100,000–200,000× *g* [30] to isolate the exosomes. Some protocols include the stage of ultracentrifugation in the sucrose density gradient [31].

Patented and commercially available sets of reagents using volume-excluding polymers are also used: for example, ExoQuick (System Biosciences, Palo Alto, CA, USA)—contains PEG 8000 kDa; Total Exosome Isolation (Thermo Fisher Scientific, Waltham, MA, USA)—includes dextrans, polyethylene glycol, polyvinyl with a molecular weight above 1000 kDa, and their combinations with other methods [32]. Different exosome isolation methods include filtration through a semipermeable membrane and isolation on immunomagnetic particles [4].

One- or two-time centrifugation of milk at low speeds before subsequent optional storage of milk plasma preliminary to exosome isolation is considered fundamental, as described in many protocols, for example, [30]. Because the composition of nucleic acids, proteins [33], and lipids [34] of milk exosomes and milk fat globule membrane (MFGM) in various articles often coincides, it cannot be excluded that storage of nondefatted milk samples leads to contamination of exosome preparations with MFGM. Refrigeration of nondefatted milk samples may lead to exosomes sticking to MFGM, resulting in further MFGM contamination of the exosome samples obtained.

Exosome preparations isolated by centrifugation and ultracentrifugation from blood plasma, urine, milk, lacrimal, and cell culture fluid usually contain “nonvesicles” with 20–100 nm diameter. These “nonvesicles” are morphologically attributed to intermediate- and low-density lipoproteins (20–40 nm) and very low-density (40–100 nm) lipoproteins [35]. These have a different structure according to transmission electron microscopy and should not be confused with exosomes.

The possibility of exosome isolation using gel filtration after or instead of ultracentrifugation [36] has been shown for human [37], bovine [38], and horse milk. Because the gel filtration allows us to effectively separate sample volumes that do not exceed 1–5% of the column volume, this method is not suitable for isolating exosomes directly from large amounts of milk. Gel filtration allows purifying the sample from co-isolating proteins in preparations obtained after ultrafiltration and/or ultracentrifugation. Our data suggest that sequential centrifugations and ultracentrifugations followed by the ultrafiltration throw 0.22 µm filter are not enough to get purified exosomes [39]. Moreover, gel filtration of sediments obtained by ultracentrifugation results in exosome preparations without admixtures of co-isolating proteins [11].

## 3. Bioactive Compounds of Milk Exosomes

The biochemical components of milk exosomes—proteins, lipids, and nucleic acids—can significantly affect therapeutic molecule delivery. In this regard, a detailed analysis of the content of these molecules in milk exosomes, also called “exosomics” (by analogy with genomics, proteomics, and other -omics technologies), is required [40].

### 3.1. Milk Exosome Proteins

The articles devoted to proteomic analysis of milk exosomes using modern highly sensitive methods describe dozens, hundreds, and even thousands of proteins and peptides, such as 115 [41], 571 [42], 2107 [43] or 2698 [44] individual proteins, and isoforms and 719 peptides [45]. According to the critical article [25], these numbers are greatly exaggerated, most likely, due to the attribution of co-isolating milk proteins with exosomes [39,46].

The milk exosomes contain several proteins that participate in their formation: Testilin controls membrane fusion [5], Rab GTPase interacts with cytoskeleton proteins [47], and Alix and Tsg101 are involved in endocytosis [48]. Moreover, milk exosomes contain proteins that determine their biological functions, namely, transport of microRNA and adhesion to the target cells (tetraspanins CD9, CD63, CD81) [49,50]. Functional activity of milk exosomal enzymes—fatty acid synthase, xanthine dehydrogenase, proteases, ADP-ribosylation factor—is not yet evident [39]. Milk exosomes also contain cytoskeleton proteins—actin, tubulin, cofilin, heat shock proteins, and molecules—involved in signal transduction, for example, in the Wnt signaling pathway [51,52]. Nuclear, mitochondrial, reticulum, and Golgi proteins cannot be an intrinsic part of exosomes as they are not transported to endosomes [53]. In any case, because the exosomes originate from an independent cellular compartment, their protein composition is not accidental.

Previously, we proposed an original method for isolating exosomes, which allows us to obtain preparations that practically do not contain co-isolating proteins from two or more liters of human of horse milk [11,24]. According to these results, horse milk exosomes contain mainly actin, butyrophilin, β-lactoglobulin, lactadherin, lactoferrin, and xanthine dehydrogenase, as well as numerous peptides (see Table 1).

These results correspond to the literature data obtained using highly sensitive proteomic methods [39], according to which butyrophilin, lactadhedrin, and xanthine dehydrogenase are the specific markers of milk exosomes. Because α-, β-, and κ-caseins and ribosomal proteins cannot be present in exosome preparations according to incompatible secretion mechanisms [40,59], these proteins are not intrinsic components and may be considered as “negative markers” of milk exosomes (fractions containing these protein markers are not exosomes or contain co-isolated protein impurities).

### 3.2. Nucleic Acids of Milk Exosomes

Studies of the past decade have shown the content of various noncoding RNA—microRNA, long noncoding RNA, circular RNA [60]—in bovine [61], human [62], panda [12], porcine [7], and rat [63] milk. Exosomal RNAs are stable at low intestinal pH values and in the presence of RNases [61,64]. Milk mRNAs are mainly concentrated in exosomes, while microRNAs are concentrated in exosomes and the supernatant [65].

The microRNA contents of milk exosomes are analyzed by high throughput sequencing (NGS) and microarray technology. Analysis of global expression profiles using microarrays revealed 79 different microRNAs (miRNAs) in the exosomal fraction and 91 in the supernatant after ultracentrifugation of bovine milk, and 39 miRNAs were common for both fractions. Further studies have shown that the expression level of these microRNAs is significantly higher in the exosomal portion compared to the supernatant [65]. Some 491 miRNAs were described in porcine milk exosomes, including 176 known miRNAs and 315 new mature miRNAs. Analysis of the gene ontology of these microRNAs showed that most of them are targeting genes related to transcriptional, immune, and metabolic processes [66]. In a study of the microRNA profile in exosomes of human milk during type I diabetes mellitus, 631 exosomal microRNAs were identified, including immune-related ones such as hsa-let-7c, hsa-miR-21, hsa-miR-34a, hsa-miR- 146b, and hsa-miR-200b. Differential expression was described for nine miRNAs in healthy and diabetic groups [67]. MircroRNAs, highly represented in milk exosomes, are reviewed in Table 2.

Different microRNA of milk and milk exosomes usually have several targets and are shown to be important in processes of fetal development, cell proliferation, pregnancy, immune system development, inflammation, sugar metabolism, insulin resistance [1,71], and many others (Table 3).

Literature data provide information on the content of 16,304 different mRNAs in porcine milk exosomes [42] and up to 19,230 mRNAs in bovine milk exosomes [65]. As with thousands of proteins, it is difficult to imagine how these thousands of mRNAs can fit in a 40–100 nm vesicle [25]. Interestingly, 18S and 28S ribosomal RNAs are practically absent in the milk-derived exosomes of human [49] and porcine milk [7], which corresponds to the current knowledge biogenesis of these vesicles.

We assume that one should be critical of information about the content of thousands of microRNA molecules and, especially, mRNA in milk exosomes. The mechanism by which these nucleic acids can co-isolate with vesicles is not as apparent as for proteins. The potential interaction of nucleic acids in milk exosomes (especially the anti-inflammatory effect and weakening of the immune response [1]) should also be taken into account when planning experiments on the delivery of therapeutic nucleic acids to cells.

### 3.3. Lipids of Milk Exosomes

Exosomes are vesicles rounded by the lipid bilayer containing proteins directed to the extracellular space. In this regard, the delivery of pharmacologically significant compounds is possible both inside and outside exosomes, both for hydrophilic (bound to surface proteins) and hydrophobic (as part of the lipid bilayer) molecules. The exosomal membrane is known to contain cholesterol, phosphatidylcholine, phosphatidylinositol, sphingomyelin, and ceramides; most phosphatidylcholine and sphingolipids has been shown to be located in the outer layer of the exosomal membrane [77]. The thickness of the lipid bilayer is at least 5 nm, so the exosome’s minimum diameter cannot be less than 40 nm. Depending on the source and the methods used, the composition of exosome lipids in milk differs from one source to another. A detailed overview of exosome lipids is given in [77,78].

The problem of exosome preparation’s contamination with lipid droplets, lipoproteins, and other particles [78] during their isolation by centrifugation and ultracentrifugation limits the study of intrinsic exosomal lipids. Exosomes isolated from cell cultures contain arachidonic acid, prostaglandins, enzymes of fatty acid metabolism [55], and leukotrienes [79]. This data indicates the need for further study of lipids in highly purified exosome preparations.

Liposomes have been widely used as targeted delivery vehicles for over 40 years. Liposome membranes can be modified to improve the targeting properties with monoclonal antibodies, aptamers, proteins, and small molecules [26]. A promising way of changing the membrane of natural exosomes is its fusion to liposomes with artificially functionalized lipids [4]. Polymer nanoparticles might have higher stability compared to liposomes, but their biocompatibility and safety in the case of long-term administration remain an unresolved problem [80]. Natural lipid components of milk exosomes are more compatible and less toxic than artificial liposomes for use as drug delivery vehicles [81].

To increase the efficiency of drug delivery by milk exosomes, a more intensive study of the lipid composition of exosomes, proteins that are located in the lipid membrane, is required. Modification of exosomal surface for targeted delivery and prolonged half-life in the body is an essential task for biopharmacology and biomedicine [82].

## 4. Milk Exosomes for Drug Delivery in Cancer Therapy

The use of antitumor drugs on the surface of natural biologically active structures, such as proteins, significantly increases the therapy’s biological availability and effectiveness [83]. Recently, much attention has been paid to the ability of exosomes to overcome physiological barriers, including the blood–brain barrier [84,85]. In this regard, many laboratories have attempted to use exosomes isolated from various cell lines for targeted drug delivery [86,87]. Despite the remarkable results shown, there are significant challenges in the development of exosomes as therapeutic products. The collection of exosomes from cell lines and/or patient samples is not compatible with commercial pharmaceutical production, and the protein component of exosomes is likely to trigger immune responses when administered systemically [85,88]. These can be overcome by using easily accessible mass sources of exosomes while solving immune responses. Studies have revealed that exosomes can be obtained from bovine milk in scalable quantities [89]; at the same time, the introduction of milk exosomes did not cause systemic toxicity and anaphylactic effect in mouse models [90]. Nonloaded camel milk exosomes contributed to a significant reduction in the progression of breast tumors, as evidenced by increased markers of apoptotic processes as well as oxidative stress reduction and downregulation of several genes associated with inflammatory processes and induction of the immune response [91]. Because the widespread consumption of milk and dairy products probably contributed to the development of human immune tolerance to milk proteins, milk exosomes can be used as delivery systems for oral drug administration [92].

In 2016, the targeting of milk exosomes to a malignant tumor was shown for the first time. Drug-loaded exosomes showed higher efficacy than free drugs in the treatment of cell cultures and xenografts of lung tumors in vivo. Milk exosomes showed interspecific tolerance without any adverse immune and inflammatory responses. Thus, the versatility of milk exosomes concerning the carrying load and the ability to reach the targeted tumor was shown [18]. In the study of milk exosomes for the delivery of the chemotherapeutic agent paclitaxel, a slight increase in the size of loaded exosomes was demonstrated, explained by the authors as partial inclusion of paclitaxel in the lipid bilayer of the membrane. At the same time, exosomes containing the drug were stable in simulated gastrointestinal conditions, highlighting their suitability for oral drug delivery [17].

Milk exosomes are used to deliver curcumin as a potential antitumor agent [93]. Exosomes significantly improved stability, solubility, and bioavailability of curcumin in adverse conditions of the digestive tract compared to free curcumin [94]. However, in a later study, it was demonstrated that curcumin loaded into exosomes derived from intestinal epithelial cells was much more effectively absorbed by colorectal adenocarcinoma cells than curcumin in milk exosomes [95]. Celastrol, a plant-derived triterpenoid, has demonstrated anti-inflammatory, antiproliferative, and antitumor activity [96,97]. However, the use of this drug is highly restricted due to its low bioavailability and toxicity. Milk exosomes, loaded with celastrol, showed high antitumor activity compared to free celastrol on the xenograft model of nonsmall cell lung cancer. Simultaneously, celastrol delivered via milk exosomes did not show gross systemic toxicity [98]. Loading preparations based on berry anthocyanins into milk exosomes [99] contributed to a significant increase in the antiproliferative activity of the drug in the experiment with cisplatin-resistant ovarian cancer cell lines [100], which also emphasizes the prospects of exosomal encapsulation when using various antitumor agents.

One of the problems of milk exosome application for targeted drug delivery is the lack of specificity to recipient cells. It has been shown that milk exosomes are absorbed from the gut as intact particles that can be modified by ligands to promote retention in target tissues [101]. Like liposome-based drug delivery systems, milk exosome-based vectors can be equipped with particular ligands to bind tumor-specific receptors [102]. The CD44 receptor, with specific ligand hyaluronan, is often overexpressed on the surface of various types of cancer cells [103]. Equipping the lipid membrane of milk exosomes with hyaluronan molecules enabled targeted delivery of the cytostatic agent doxorubicin to cells expressing CD44 [102]. The addition of folic acid as a tumor receptor ligand to milk exosomes loaded with a chemotherapeutic agent significantly enhanced tumor cell growth inhibition in a mouse xenograft model [18]. Bovine milk exosomes in vitro activated CD69 (an early marker of lymphocyte proliferation) on the normal killer (NK) cells. The stimulation of γ-interferon (IFN) production by NK-cells and T-lymphocytes after co-activation with milk exosomes and interleukins 2 and 12 indicates the possible undesirable enhancement of the inflammatory process [104].

Among current trials registered on clinicaltrials.gov, none are using milk exosomes as drug delivery systems. Nevertheless, studies demonstrate that the use of milk exosomes for the delivery of antitumor drugs contributes to a significant increase in their effectiveness and reduces the toxicity of the therapy, which opens up excellent prospects for the application of milk exosomes for oral delivery of therapeutic agents [18]. Molecules delivered with milk exosomes are reviewed in Table 4.

## 5. Biological Activity of Milk Exosome Nucleic Acids and Delivery of Nucleic Acids to Cancer Cells

Cell lines of acute leukemia and colon cancer capture breast milk exosomes and their RNA contents. MicroRNAs from both the fat layer and the supernatant were detected in these cells. An increase in the expression level of miR-148a-3p and a decrease in the DNA methyltransferase I (DNMT1) gene, which is its direct target, were also detected [106]. These facts suggest that RNAs contained in milk exosomes can directly or indirectly affect the growth and development of cancer cells. Men who consumed more than three daily norms of high-fat bovine milk showed an increased risk of prostate cancer [107]. Milk exosomes supplemented to prostate cancer cell cultures increased their proliferative activity by 30% [108]. These effects are most likely related to the microRNA contents of human [109], porcine [66], and bovine [110] milk that can significantly reduce the expression level of the TP53 gene, with negative regulation one of the typical events in carcinogenesis. MicroRNA, presented in human milk exosomes, may control up to 9074 genes [109,111].

The main bovine milk microRNA miR-21 was found to target p57Kip2, a factor that inhibits cyclin-dependent kinases in prostate cancer cells [112]. Elevated levels of miR-21 were found in the serum and tumor tissue of breast cancer patients [113,114]. P57kip2 knockdown enhances the proliferative phenotype induced by tumor-associated mutant variants of phosphoinositide 3-kinase. It releases mammary epithelial acinus from cell arrest during morphogenesis [115], suggesting a potential effect of mammary miR-21 on tumor growth processes. Long noncoding RNA of porcine milk exosomes may interact with miRNA related to cell proliferation, and circular RNA may target miRNA associated with intestinal cells [60]. A recent paper showed that miRNAs from bovine and porcine milk were adsorbed in piglet intestine and transported to the blood serum with different efficiencies [92].

Exosomes stimulate the proliferation of intestinal epithelium cells of healthy but not tumor cultures. Incubation of healthy cells with milk exosomes results in a decrease of the phosphatase and tensin homolog (PTEN) expression level, which is the main target of miR-148a, the content of which has also been shown for milk exosomes [65,74]. In miR-148a knockout cells, milk exosomes suppress the proliferation and expression of DNMT1 [116]. Thus, studies have shown the content of many different microRNAs in milk exosomes and their bioavailability and ability to modify the expression of multiple genes, including those associated with the occurrence and growth of malignant tumors.

The ability of exosomes to transfer mRNA and microRNA between cells and subsequently mediate changes in target gene expression in recipient cells also highlights their potential for delivering exogenous small interfering RNAs. Small interfering RNAs (siRNAs) are a potential generation of new therapeutics. With increasing knowledge on the endogenous RNA interference, the possibility of using siRNA as nucleic acid-based drugs for many diseases, including different types of cancer, is growing [117]. Despite the tremendous therapeutic potential, siRNAs are hindered by the lack of effective ways to deliver them to the cell. Chemical modification of small interfering RNAs for their delivery was unsuccessful and, in some cases, led to the loss of their biological activity [118,119] and the occurrence of toxic effects [120]. Polymer nanoparticles, lipids and liposomes, peptides, and synthetic nanocarriers were considered alternative options for delivering siRNA to cells; the most developed delivery system is based on liposomes [121]. However, these methods never solved the problem of nonspecific targeting and immune response. Bovine milk exosomes can deliver endogenous RNA load to recipient cells, and it remains stable, resisting degradation [105]. The dose-dependent antiproliferative activity of exosomes loaded with siRNAs interfering with the mutant allele of the KRAS gene in cell lines and mouse xenografts of the lung cancer were demonstrated [20].

## 6. Conclusions

The development of biocompatible methods of delivering therapeutically significant drugs in vitro and in vivo is essential for molecular pharmacology. Exosomes of horse, goat, and sheep milk are free of prion proteins [16] and have excellent prospects for practical application. Still, some problems limiting the use of milk exosomes in therapy remain unresolved: insufficient data on both proteins and nucleic acids that are an intrinsic part of exosomes and which are co-isolated during purification; the specificity of milk exosomes to various cells and tissues; and the development of methods for modifying the surface of milk exosomes for target delivery.

Delivery by artificially produced liposomes is often compared with natural exosomes, and various reasons are given pro et contra use of natural vesicles. The main advantage of milk exosomes for drug delivery compared to liposomes and other artificial nanoparticles is higher biocompatibility, namely, lower immunogenicity and cytotoxicity [122], due to the chemical composition of exosomes similar to cell membranes [80]. Milk exosomes can deliver both hydrophilic and hydrophobic molecules; the isolation from milk is scalable and cheaper compared with isolation from blood plasma or culture fluid.

Biologically active molecules carried by milk exosomes remain stable even in the digestive tract’s harsh conditions, making it possible to use milk exosomes for oral drug delivery [18,101]. The described diversity of various miRNAs in the composition of milk exosomes and their influence on the implementation of genetic information in recipient cells opens up a vast potential for exosomes in gene therapy of diseases, especially oncological ones. The lipid composition of milk exosomes, which plays an essential role in the delivery of drugs and biologically active molecules, is also insufficiently studied. The facts above indicate the relevance for further study of the biochemical composition of milk exosomes.

## Figures and Tables

**Table 1 ijms-21-06646-t001:** Highly represented proteins of milk exosomes.

Highly Presented Proteins	Number of Proteins Described in a Paper	Source of Milk Exosomes, Ref	Method of Analysis
Butyrophilin, κ-casein, lactadherin, xanthine dehydrogenase	94	Bovine [54]	LC-MS/MS of tryptic hydrolysates
Angiogenin-1, lactoferrin, lactoperoxidase sulfhydryl oxidase	920	Bovine [55]	LC-MS/MS of tryptic hydrolysates with iTRAQ
Butyrophilin, CD36, complement component 3, fatty acid synthase, lactadherin, lactotransferrin, low-density lipoprotein receptor-related protein 2, polymeric immunoglobulin receptor, xanthine dehydrogenase	1372	Bovine [56]	LC-MS/MS of tryptic hydrolysates
α-casein, butyrophilin, fatty acid-binding protein, lactadherin, α-lactalbumin, β-lactoglobulin, xanthine dehydrogenase	1879	Bovine [57]	LC-MS/MS of tryptic hydrolysates
Adipophilin, butyrophilin, lactadherin, xanthine oxidase	2107	Bovine [43]	LC-MS/MS of tryptic hydrolysates
Butyrophilin, lactadherin, fatty acid synthase, xanthine dehydrogenase	2299	Bovine [58]	LC-MS/MS of tryptic hydrolysates with iTRAQ
Actin, butyrophilin, lactadherin, lactoferrin, β-lactoglobulin	8	Horse [11]	MALDI-TOF-MS/MS of tryptic hydrolysates after 2D-Electrophoresis
CD36, α-enolase, fatty acid synthase, lactadherin, lactotransferrin, polymeric-Ig-receptor, Rab GDP dissociation inhibitor, syntenin-1, xanthine dehydrogenase	73	Human [5]	LC-MS/MS of tryptic hydrolysates
β-Casein, lactoferrin, serum albumin polymeric Ig receptor, tenascin, xanthine dehydrogenase	115	Human [41]	LC-MS/MS of tryptic hydrolysates
Annexins, CD9 CD63, CD81, flotillin, G-protein subunits, lactadherin, Rab, Ras-related proteins, syntenin	2698	Human [44]	LC-MS/MS of tryptic hydrolysates
Albumin, ceruloplasmin, complement C, α-glucosidase, fibronectin, lactotransferrin, thrombospondin	571	Porcine [42]	LC-MS/MS of tryptic hydrolysates after SDS PAGE

**Table 2 ijms-21-06646-t002:** Highly represented microRNA of milk exosomes.

Highly Presented microRNA	Number of microRNAs Described in a Paper	Source of Milk Exosomes, Ref	Method of Analysis
2478, 1777b, 1777a, let-7b, 1224, 2412, 2305, let-7a, 200c, 141	79	Bovine [65]	Microarray
148a, let-7c, let-7a-5p, 26a, let-7f, 30a-5p, 30d	372	Buffalo [68]	RNA seq
30d-5p, let-7b-5p, let-7a-5p, 125a-5p, 21-5p, 423-5p, let-7g-5p, let-7f-5p, 30a-5p, 146b-5p	219	Human [69]	RNA seq
22-3p, 148a-3p, 141-3p, 181a-5p, 320a, 378a-3p, 30d-5p, 30a-5p, 26a-5p, 191-5p	308	Human ^1^ [66]	RNA seq
let-7a-5p, 148a-3p, 146b-5p, let-7f-5p, let-7g-5p, 21-5p, 26a-5p, 30d-5p	631	Human [70]	RNA seq
148a-3p, 30b-5p, let-7f-5p, 146b-5p, 29a-3p, let-7a-5p, 141-3p, 182-5p, 200a-3p, 378-3p	602	Human [67]	RNA seq
let-7b-5p, 92a-3p, 148a-3p, 30a-5p, let-7a-5p, 181a-5p, let-7i-5p, let-7f-1/2-5p, let-7g-5, 200a-3p	1191	Panda [12]	RNA seq
148a-3p, 182-5p, 200c-3p, 25-3p, 30a-5p, 30d-5p, 574-3p	234	Porcine [7]	RNA seq
148a, let-7b, let-7a, 21, let-7c, let-7i, 26a, let-7f, 125b, 143	84	Sheep [14]	RNA seq

^1^ Preterm delivery.

**Table 3 ijms-21-06646-t003:** Genes regulated by a highly abundant microRNA of milk exosomes.

microRNA(s)	Gene(s) Targeting with microRNA	Biological Function(s) of the microRNA Targets, Ref
22-3p	Transcription factor 7 of Wnt pathway	Regulation of gluconeogenesis, insulin resistance [72]
25-3p	KLF4 (Krüppel-like factor 4)	Development of the immune system [7]
30a-5p	P53, DRP1 (dynamin-related protein 1), GALNT7 (GalNAc transferase 7)	Mitochondrial fission, cellular invasion, immunosuppression, synthesis of interleukin (IL)-10 [7]
30d-5p	GalNAc transferases	Inflammatory processes [73]
148-3p	NF-κB (transcription factor)	Decrease of the immune response [74]
148a ^1^	DNMT1	Epigenetic regulation [71]
let-7 family	Insulin-PI3K-mTOR signaling pathway	Glucose tolerance and insulin sensitivity [75]
let-7a-5p let-7b-5p	TRIM71, IL6-induced signal activation of transcription,	Stem cells proliferation, fetal development [74], activation of metalloproteinases [76]

^1^ The most expressed miRNA of human milk.

**Table 4 ijms-21-06646-t004:** Biologically and therapeutically significant molecules delivered with milk exosomes.

Molecule Delivered	Solubility in Water	Source of Milk Exosomes, Ref	Cell Lines Used for Delivery
Anthocyanins	Soluble	Bovine [99,100]	A549 ^1^, H1299 ^1^, MDA-MB-231 ^2^, MCF7 ^2^, PANC1 ^3^, Mia PaCa2 ^3^, PC3 ^4^, DU145 ^4^, HCT116 ^5^, OVCA432 ^6^, OVCA433 ^6^, A2780 ^6^, A2780/CP70 ^6^
Celastrol	Insoluble	Bovine [98]	A549 ^1^, H1299 ^1^
Curcumin	Insoluble	Bovine [94,95,98]	Caco-2 ^5^, H1299 ^1^, A549 ^2^, HeLa ^7^, MDA-MB-231 ^2^, T47D ^2^
Docetaxel	Insoluble	Bovine [18]	A549 ^1^, H1299 ^1^, MB-231 ^2^, T47D ^2^, Beas-2B ^8^
Doxorubicin	Soluble	Bovine [102]	A549 ^1^, MDA-MB-231 ^2^, MCF-7 ^2^, HEK293 ^9^
Paclitaxel	Insoluble	Bovine [17,18]	A549 ^1^, H1299 ^1^, MB-231 ^2^, T47D ^2^, Beas-2B ^8^
siRNA	Soluble	Bovine [20,105]	A549 ^1^, H1299 ^1^, MDA-MB-231 ^2^, MCF7 ^2^, PANC1 ^3^, Mia PaCa2 ^3^, Caco-2 ^5^, A2780 ^6^
Withaferin A	Insoluble	Bovine [18]	A549 ^1^, H1299 ^1^, MB-231 ^2^, T47D ^2^, Beas-2B ^8^

^1^ lung cancer, ^2^ breast cancer, ^3^ pancreatic cancer, ^4^ prostate cancer, ^5^ colon cancer, ^6^ ovarian cancer, ^7^ cervical cancer, ^8^ normal bronchial epithelium, ^9^ normal kidney embryonic cells.

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
