# Peer review of "Milk Exosomes: Perspective Agents for Anticancer Drug Delivery"

_ijms, 2020, doi:10.3390/ijms21186646_

Round 1
Reviewer 1 Report
This review is well-written and well-constructed. However, the reviewer would like to ask the authors’ opinions about the following questions:
- Exosomes contain numerous proteins, mRNAs, and miRNAs. Among them, some components have negative influences on delivered cells. Therefore, I disagree with the usage of whole exosomes for DDS. The authors had better describe how to use exosomes for DDS. If the authors remove the contents, how do the authors do?
- Although exosomes obtained from horse, goat, and sheep milk are free of prion, their antigenicity is different from human. Because of species specificity, I think that we have to use human milk exosomes for the clinical settings. How do the authors collect human milk exosomes?
Author Response
1) Exosomes contain numerous proteins, mRNAs, and miRNAs. Among them, some components have negative influences on delivered cells. Therefore, I disagree with the usage of whole exosomes for DDS. The authors had better describe how to use exosomes for DDS. If the authors remove the contents, how do the authors do?
We added some discussion (please see lines 55-66)
2) Although exosomes obtained from horse, goat, and sheep milk are free of prion, their antigenicity is different from human. Because of species specificity, I think that we have to use human milk exosomes for the clinical settings. How do the authors collect human milk exosomes?
We agree that exosomes isolated from human milk might be the best for human treatment. However, this source is also have some limitations due to the fact that, for example, HIV is easilyt transmitted through milk. Also, a problem may be the fact that today we are not familiar with the donors of human milk who are able to produce it on an industrial scale. According to our results, higly purified preparations of milk exosomes (from human, horse, bovine milk) do not contain or practically do not contain allergenic proteins such as caseins, lactalbumin, and lactoglobulin (two articles are being prepared for publication). In this regard, we hope such higly purified exosomes isolated from non-human milk may be successfully used for the delivery of drugs and nucleic acids into cells.
We added the general protocol isolation of exosomes from milk (see lines 72-78) and particular details of our protocol (see lines 99-106).
Reviewer 2 Report
The article titled “Milk exosomes - perspective agents for anti-cancer drug deliver” is a well written review on a subject of current interest. The article discusses isolation of exosomes, their components (proteins, nucleic acid and lipids etc) and sheds lights on delivery of small therapeutic drugs and large molecules. References cited in the article are up to date. There are minor comments that the authors should address before it is accepted for publication.
- Authors should consider tabulating the various studies with delivery of small and large molecules (from section 4 and 5).
- Since the review is specifically on drug delivery, it would be appropriate to include the following articles in section 4.
- Cancer Lett. 2017 May 1;393:94-102 - Exosomal formulation of anthocyanidins against multiple cancer types.
- AAPS J. 2017 Nov;19(6):1691-1702 - Exosomes for the enhanced tissue bioavailability and efficacy of curcumin.
- Line 158 - delete cholesterol as it appears twice.
- Line 169-171 can be deleted. It appears non-relevant and non-connected.
- Line 213 – mention anti-inflammatory activity as reference 94 specifically talk about that.
Author Response
Authors should consider tabulating the various studies with delivery of small and large molecules (from section 4 and 5).
We added a table, reviewing the molecules, delivered with milk exosomes, please see Table 4 (lines 263-265)
Since the review is specifically on drug delivery, it would be appropriate to include the following articles in section 4.
1) Cancer Lett. 2017 May 1;393:94-102 - Exosomal formulation of anthocyanidins against multiple cancer types.
2) AAPS J. 2017 Nov;19(6):1691-1702 - Exosomes for the enhanced tissue bioavailability and efficacy of curcumin.
Please, see refs 94, 100
Line 158 - delete cholesterol as it appears twice.
The second mention of cholesterol removed
Line 169-171 can be deleted. It appears non-relevant and non-connected.
The non-relevant phrases were removed
* Line 213 – mention anti-inflammatory activity as reference 94 specifically talk about that.
We added the mention of anti-inflammatory activity, please see line 237
Reviewer 3 Report
This review provides some good information about milk exosomes, their purification and their contents. However, the title suggests that a major focus of this review will be on drug delivery for anti-cancer agents. While this topic is addressed, the majority of the review focuses on exosomal contents and some general comments about the use of miRNA. While there are many potential uses for milk exosomes, this review does not provide a justification as to why exosomes offer advantages over other nanoparticles (e.g., liposomes). Considering the cost and effort needed to isolate exosomes from milk and the potential for immune responses, it would be very helpful if a rationale is provided for pursuing milk exosomes as delivery vehicles; this is not explicitly stated and I feel that a review on this topic needs to address this issue. In addition some of the wording is awkward, e.g., section 2 describes contamination of MFGM preparations with exosomes, but I believe the focus of the review is on exosome preparations. Similarly, lines 76-77 and 112-114 need to be clarified. The mention of targeting and solving toxicity problems (lines 197-198) are vague, and it is unclear how toxicity problems can be addressed by using exosomes.
Author Response
Considering the cost and effort needed to isolate exosomes from milk and the potential for immune responses, it would be very helpful if a rationale is provided for pursuing milk exosomes as delivery vehicles; this is not explicitly stated and I feel that a review on this topic needs to address this issue.
We added some discussion, please see lines 318-322
In addition some of the wording is awkward, e.g., 73-75 section 2 describes contamination of MFGM preparations with exosomes, but I believe the focus of the review is on exosome preparations.
We clarified the text, please see lines 89-91
Similarly, lines 76-77 and
We clarified the text, please see lines 95-97
112-114 need to be clarified
We changed the text, please see lines 138-141
The mention of targeting and solving toxicity problems (lines 197-198) are vague and it is unclear how toxicity problems can be addressed by using exosomes.
We changed the text, please see line 222
Round 2
Reviewer 3 Report
The justification for using milk exosomes for delivery provided in the revised version (320-322) are biocompatibility, lower toxicity, the ability to deliver hydrophilic and hydrophobic molecules, availability, scalability, and reproducibility. Are the authors arguing that a milk exosome is more compatible and less toxic than an old-fashioned liposome? Have they considered immunogenicity? Liposomes can also deliver hydrophilic and hydrophobic molecules, and have been scaled up for commercial production. My point isn't that liposomes are so great, the point is that the review needs to tell the reader what advantages a milk exosome has over a traditional particle like a liposome. If there is no advantage, why go through the process of isolating from milk, etc.
On lines 221-222, the authors mention that milk exosomes can be used for oral drug delivery. Isn't this the main advantage of exploring them? Liposomes or other nanoparticles cannot do this efficiently; this has been compared in the literature. Are the authors proposing milk exosomes for cancer treatment via other routes of administration (non-oral)? This needs to be clarified throughout the manuscript. What do the authors mean when they talk about "directed transport"? The authors need to be specific about this instead of providing vague claims of solving "problems".
Author Response
Dear Reviewer!
Are the authors arguing that a milk exosome is more compatible and less toxic than an old-fashioned liposome?
Yes, we argue that milk exosomes are better drug delivery systems than liposomes. We tried to emphasize on that feature; please see lines 199-202, 322-328
Have they considered immunogenicity?
Please see line 325
The point is that the review needs to tell the reader what advantages a milk exosome has over a traditional particle like a liposome.
Please see lines 199-202, 322-328
On lines 221-222, the authors mention that milk exosomes can be used for oral drug delivery. Isn't this the main advantage of exploring them?
Please see lines 21, 225, 234, 264-265, 329-330, focusing on the use of milk exosomes for oral drug delivery.
Are the authors proposing milk exosomes for cancer treatment via other routes of administration (non-oral)?
We cannot exclude the possibility of using milk exosomes for the non-oral route of administration. Since the literature data available to the current time is not enough, we are not ready to discuss these possibilities.
What do the authors mean when they talk about "directed transport"? The authors need to be specific about this instead of providing vague claims of solving "problems".
We removed "directed transport" from line 20.
Sincerely, Sergey Sedykh and the co-authors